**Data Availability Statement:** All relevant data are within the paper and its Supporting Information files.

**Funding:** The author(s) received no specific funding for this work.

# Sero-prevalence and associated factors of sexually transmitted infections among youth-friendly services Attendees

**Erdachew Ambaye[1], Moges Desta Ormago[2], Musa Mohammed Ali[2]***

**1** South Nation Nationality People public health Laboratory, Hawassa, Ethiopia, **2** School of Medical Laboratory Science, Hawassa College of Medicine and Health Sciences, Hawassa University, Hawassa, Ethiopia

* ysnmss@yahoo.com

## Abstract

### Background

Worldwide, more than one million peoples acquire sexually transmitted infections (STIs). The burden of STIs and the youth awareness level on the transmission of STIs is under investigated in Sidama Regional State.

### Objective

To determine the seroprevalence of STIs such as hepatitis B surface antigen (HBsAg), Anti-hepatitis C virus (HCV) antibodies, Human Immunodeficiency virus (HIV) seroprevalence, and syphilis and to determine associated factors among youth-friendly services Attendees at selected health facilities in Hawassa city, Ethiopia.

### Methods

A multicenter cross-sectional study was conducted among 416 randomly selected youth attending youth-friendly services at selected health facilities from May to August 2021. To collect the background characteristics of participants an interviewer-administered questionnaire was used. Blood samples were collected, processed, and tested using Advanced Quality One Step rapid colloidal gold immunochromatographic assay for detection of antibodies for syphilis and hepatitis C virus infection, and hepatitis B virus surface antigen. For the diagnosis of syphilis Rapid Plasma Reagin was also used. HIV1/2 STAT PAK, HIV1/2/O ABON and HIV1/2 SD Bioline were used for testing antibodies for HIV infection. Data entry and analysis were performed using Statistical Package for the Social Sciences (SPSS) version 20.0 software. A crude and adjusted odds ratio with a 95% confidence interval (CI) was computed to identify associated factors.

### Results

The overall seroprevalence of STIs was 11.5% (48/422), with a 95% CI: (8.7−14.9). Out of the 48 positive results, the proportions of HBsAg, Anti-HCV, HIV, and syphilis were 56.3%

**Competing interests:** The authors have declared that no competing interests exist.

**Abbreviations:** STI, Sexually transmitted infection; FGAE, Family guidance association of Ethiopia; HBsAg, Hepatitis B surface antigen; HBV, Hepatitis B virus; HCV, Hepatitis C virus; HIV, Human immunodeficiency virus; SNNPR, South nation's nationalities and peoples' region.

(27/48), 27.1% (13/48), 10.4% (5/48), and 6.3% (3/48) respectively. Out of 416 participants, 17.1% responded that it is safe to have sex without using a condom. The Odds of developing STI among female participants, participants who did not identify alcohol intake as a risk factor for STIs, and those who engaged in transactional sex were (AOR = 2.989: 95% CI: 1.27, 7.02), and (AOR = 2.393, 95% CI: 1.18, 4.81) and practice of transactional sex (AOR = 5.527, 95% CI: 1.62, 18.75).

## Conclusions

STIs are common among youth-friendly services Attendee in Hawassa city The overall STI was significantly associated with sex (females), not able to identify alcohol intake as a risk factor, and practice of transactional sex. High number of (n = 289, 69.5%) participants think that the use of condoms is not effective in preventing STIs and they engage in sexual activity without using condoms.

## Background

Sexually transmitted infections (STIs) spread from one individual to another through sexual contact, most of them are easily preventable and curable [1]. There are over 30 bacterial, viral, fungal, and parasitic pathogens that have been identified to date to be transmitted sexually [2, 3].

STI is a major public health problem worldwide it does not only affect the quality of life but also causes high morbidity and mortality. Besides, syphilis may cause neurological, cardiovascular, and dermatological disease in adults and causes stillbirth, neonatal death, premature delivery, or severe disability in infants [2, 4].

Human Immunodeficiency virus (HIV), viral hepatitis, and other STIs collectively cause 2.3 million deaths each year and continue to impose a major public health burden worldwide [5]. Collectively, more than one million people are newly infected with STIs, and 4.5 million with HIV, hepatitis B, and hepatitis C, each year [5].

The epidemiology of STI varies depending on the nature of the population and the types of STI studied [6]. The prevalence of syphilis among pregnant women in Ethiopia was 2.53% [7] whereas the prevalence of HBsAg and anti-HCV in Gambella town, Ethiopia among refugees was 7.3% and 2% respectively [8]. The seroprevalence of HIV among blood donors in Ethiopia was 2.83% [9]

Globally, young populations an active working group of a society, make up only one-fourth of the sexually active population, but they acquire 50% of all STIs [10]. Moreover, young populations are prone to STIs because of a lack of awareness, unsafe sexual intercourse, and the tendency not to use condoms [11, 12]. The highest proportions of STIs are reported among the population that belongs to the 15–24 year age group. World Health Organization (WHO) estimate indicates that one in 20 young people are believed to contract STI each year [13].

As youth are an active working group of society, STIs constitute huge health and economic burden, particularly in developing countries where they account for 17% of economic losses due to ill-health [14]. STIs result in substantial productivity losses at individual and community levels [15].

In developing countries, STIs and their complications are amongst the top five disease categories for which youths and adults seek health care [16]. Sub-Saharan Africa carries a high

burden of STIs, contributing to more than 70% of the burden of infection [17]. Designing or strengthening existing prevention strategies which target the youths are essential to improve their health. This is particularly true for STIs, which, when acquired during adolescence, can affect sexual, and reproductive health later in life and, for women, the health of their babies can be affected [18].

STI disproportionately affects the health and social well-being of women and men by producing a significant impact on their reproductive potential [19]. Young women are biologically prone to STIs as compared to older women. The reason behind this is vaginal mucosa of the genital area is immature making young women vulnerable to STIs [20].

STIs can be prevented by delaying the onset of sexual activity, and abstaining from sexual intercourse until marriage occurs. If young people are already sexually active, it is important to make sure that they know how to use condoms [20].

Even though there are several studies on STIs in Ethiopia, there is scarce information on STIs among youths attending youth-friendly services. This study aimed to determine the sero-prevalence and associated factors of STIs among youth attending youth-friendly services at selected health facilities in Hawassa city. Hawassa has a total population of 523,175 people according to the 2016 report. In the city, 10 health facilities are expected to have youth centers to provide youth-friendly services; however, only 4 of them provide sexual and reproductive health services to their full capacity. The services they provide include sexual and reproductive health information and education, peer education by peer and health providers, pregnancy testing, HIV/AIDS/STI diagnosis, and management, counseling service, family planning, antenatal care, postnatal care, and infertility management.

## Methods

### Study area and design

A facility-based cross-sectional study was conducted from May 1 to August 30, 2021, among youth attending youth-friendly services at selected health facilities in Hawassa, Sidama Regional State, Ethiopia.

### Study population

All symptomatic and asymptomatic youth between the ages of 15–24 years, those who were able and willing to provide consent and assent, and those who attended youth-friendly services at youth centers during the study period in Hawassa city. These health facilities were selected based on the data obtained from the Hawassa city health department. These facilities have been providing all sexual and reproductive health services during the study period. Youth, who was diagnosed with STI and are on treatment for STI; youth who were sick and unable to participate in the interview were excluded.

### Sample size determination

The sample size was determined using single population proportion formula (N = z2 p(1-p)/d2), considering 95% confidence interval, 0.5 proportion, 5% marginal of error and 10% for non-response rate. Considering all this, the total sample size was computed to be 422. A systematic random sampling method was applied for the selection of participants. Considering a four-month study period, a total of 2102 youths were expected to visit the youth-friendly services according to the health facilities plan and the past four month's performance report. This estimate was divided by the total sample size (2102/422) to determine the sampling interval (k value), which was 4.9 (~5). The 1st participant was selected by lottery method and then every

5th client afterward was invited to participate in the study until the required sample size was achieved. The sample size was allocated for each study site proportionally (422/2102x past four months' patient flow of the site). The total allocated sample sizes for the four study sites were 258, 38, 9, and 117.

## Socio-demographic and behavioral data

The interviewer-administered structured questionnaire adapted from previous studies was used to collect socio-demographic and behavioral data [21]. Data collected include age, sex, marital status, family residence, monthly income, knowledge about STIs, substance use, alcohol use, practice and knowledge related to sex, and previous history of STIs.

## Sample collection

A 5ml whole blood specimen was collected using a syringe from a vein and shipped with a triple package system to South nation's nationalities and peoples' region (SNNPR) Public Health Institute Laboratory. Serum was prepared by centrifuging a blood specimen for 5 minutes at 3000 Revolution per minute (RPM) and transferred to a Nunc tube and stored at −20˚C until they were tested.

The presence of HBsAg in the serum was detected using an advanced quality one-step HBsAg rapid immunochromatographic assay according to the manufacturer's manual (Intec PRODUCTS, INC. XIAMEN, P.R. China).

The presence of antibodies to Hepatitis C virus (HCV) was detected by advanced quality one-step rapid anti-HCV colloidal gold immunochromatographic assay according to the manufacturer's manual (Intec PRODUCTS, INC. XIAMEN, P.R.China).

The presence of antibodies to syphilis antigen in human serum was detected by using Rapid Plasma Reagin (RPR) and Rapid Syphilis test one step anti-TP according to the manufacturer's manual immunochromatographic assay (Intec PRODUCTS, INC. (XIAMEN, P.R.China).

For HIV diagnosis, we have used the national HIV rapid diagnostic test algorithm to determine antibodies. Initially, HIV infection was screened using STAT PAK (Chembio diagnostic, INC, Medfold, Newyork). HIV-positive samples were re-tested with the 2nd test, ABON (Abon Biopharm (Hangzhou) co., Ltd. P.R.China). HIV-positive samples by ABON were re-tested with the 3rd test SD Bioline (SD STANDARD DIAGNOSTICS, INC. Republic of Korea).

## Data quality assurance

A structured questionnaire prepared in English was translated into Amharic language and then translated back to English to check for its consistency. The interview was conducted using the Amharic language at selected health facilities by trained health professionals. To ensure the quality of data, before initiation of the study, data collectors (laboratory professionals and nurses) were trained. A structured questionnaire was pretested among a population representing 5% of the total sample size in Alamura health center before initiating the study. After preliminary data collection during the pre-test, the questionnaire was assessed by the principal investigator for its clarity, logical flow, length, culture sensitivity, and completeness and the necessary correction was made. Close supervision was made during data collection, laboratory investigation, data entry, and analysis. Every day, collected data were checked and reviewed for completeness and clarity.

Standard operating procedures of the SNPPR public health laboratory were followed during sample collection, handling, transporting, and processing. Expiry date and lot number of each test kit and reagent were checked before use. The presence or absence of a band/line in the

control area within the device itself for HIV, HBV, HCV, and syphilis was checked before interpreting the result. Control samples (both positive and negative controls) provided with the test kit were used to verify whether the procedure is working properly or not. External quality control materials from the blood bank and public health institute laboratory were used to verify test kits. The test kits used in this study were selected based on their sensitivity and specificity (ranging from 95–100%).

## Data analysis

Data were checked for completeness before entering into a computer. Data entry and analysis were performed using SPSS version 20. Bivariable binary logistic regression was used for preliminary assessment of the association between the seroprevalence of STIs and independent variables. A multivariable binary logistic regression model was used for variables with a p-value less than 0.25 in bivariable binary logistic regression analysis. Odds ratio, 95% confidence interval, and p-value were calculated to determine the strength of association. A $p < 0.05$ was taken as a cut point to declare statistical significance.

## Operational definition

STIs are defined as a positive status either for a single or a combination of infections specific to HIV, HBV, HCV, and syphilis.

Transactional sex: heterosexual intercourse in exchange for money and/or material goods.

## Ethical consideration

Ethical approval was obtained from Hawassa University College of medicine and health sciences Institutional Review Board (IRB) (Reference number: IRB/209/13). Written informed consent was obtained before the recruitment of study participants. For minors, written informed consent from parents/guardians and assent from minors were obtained before data collection. The objectives, purpose, benefit, risk of the study, and their role in the study were briefed to them. To ensure confidentiality; participants' data was linked to a code and if the test showed positive results, health education was provided to the participants, and the laboratory findings were communicated to attending health care workers in respective health facilities for management.

## Results

### Socio-demographic characteristics

A total of 416 participants aged 15–24 years with mean age (±SD), 19.8(±2.69) were interviewed with a response rate of 98.6% (416/422). More than half of the study participants 60.6% (252/416) belong to the 20–24 year age group; 69.2% (288/416) were females and 80.5% (335/416) were urban dwellers. Close to 30% of participants work in private organizations whereas the majority of them 48.6% (202/416) were students. Of the total participants, 33.2% (138/416) completed elementary school and 42.3% (176/416) completed secondary education (**Table 1**).

### Knowledge and behavior related characteristics

The majority of participants 84.9% (353/416) reported that they knew the routes of STIs transmission; they mentioned sexual intercourse as a main route of transmission. Furthermore, 69.5% (289/416) participants think that the use of condoms is an effective way of preventing STIs, and 71.6% (298/416) reported that STIs including HIV cannot be transmitted through hugging and shaking hands. More than half (57.5%) reported that STIs can increase the risk of

**Table 1. Socio-demographic characteristics of youth-friendly services Attendees at selected health facilities in Hawassa city, Sidama Regional State, Ethiopia, May 1 to August 30, 2021 (N = 416).**

| Variables | Category | Total N (%) |
|---|---|---|
| Age in years | 15–19 | 164 (39.4) |
| | 20–24 | 252 (60.6) |
| Previous place of residence | Rural | 81 (19.5) |
| | Urban | 335 (80.5) |
| Marital Status | Married | 90 (21.7) |
| | Single | 326 (78.3) |
| Occupation | Government employee | 14 (3.4) |
| | Private Organization | 122 (29.3) |
| | Unemployed | 32 (7.7) |
| | Merchant | 39 (9.4) |
| | Student | 202 (48.6) |
| | Others | 7 (1.7) |
| Educational Status | No formal education | 4 (1.0) |
| | Read and write | 9 (2.2) |
| | Elementary | 138 (33.2) |
| | Secondary | 176 (42.3) |
| | Diploma and above | 89 (21.4) |
| Monthly income of house hold in Ethiopian Birr | No income | 191 (45.9) |
| | < 1500.00 | 141 (33.9) |
| | 1500.00–2500.00 | 63 (15.1) |
| | > 2500.00 | 21 (5.0) |

HIV infection and 61.8% (257/416) replied that it is not safe to have sex without a condom (Tables 2 and 3).

## Seroprevalence of STIs and its predictors

Prevalence of each infection was as follows: HBsAg 6.5% (27/416), Anti-HCV 3.1% (13/416), Syphilis 0.7% (3/416) and HIV 1.2% (5/416). The overall seroprevalence of STIs was 11.5% (48/416), with 95% CI: (8.7–14.9). Out of the 48 positives, the proportion of each tests are presented as follows: HBsAg (n = 27; 56.3%), Anti-HCV (n = 13; 27.1%), HIV (n = 5; 10.4%), and RPR (n = 3; 6.3%). One of the participants was coinfected with HBV and HCV and another participant was also coinfected with HBV and HIV.

In bivariable binary logistic regression analysis, the following factors had a $p<0.25$ and were selected for multivariable analysis: age, sex, place of previous residence, marital status, identification of alcohol intake as a risk factor for STIs, identification of multiple sexual partners as a risk factor for STIs, involvement in transactional sex, and ever had sex without using a condom. In multivariable analysis, being female (AOR = 2.989, 95% CI: 1.272, 7.024) was associated with higher odds of STIs. Participants who did not identify alcohol intake as a risk factor for STIs were associated with higher odds of STIs (AOR = 2.393, 95% CI: 1.188, 4.817) (Table 4).

## Discussion

In the current study, the overall seroprevalence of STIs among youth was 48(11.5%), with a 95% CI: (8.7–14.9). Out of the 48 positive results, the highest proportion was observed for HBsAg (56.3%), followed by HCV (27.1%), HIV (10.4%), and syphilis (6.3%). The overall STI

**Table 2. Knowledge/beliefs related characteristics among males and females youth-friendly services Attendees at selected health facilities in Hawassa city, Sidama Regional State, Ethiopia, May 1 to August 30, 2021 (N = 416).**

| Variables | Category | Total N (%) | Sex | | p-value |
|---|---|---|---|---|---|
| | | | Male n(%) | Female n(%) | |
| Routes of transmission for STI | Sexual intercourse | 353 (84.9) | 109(26.2) | 244(58.7) | 0.821 |
| | Blood transfusion | 5 (1.2) | 1(0.2) | 4(1.0) | |
| | Sharing injection needles | 14 (3.4) | 3(0.7) | 11(2.6) | |
| | Sharing food/drink | 9 (2.2) | 4(1.0) | 5(1.2) | |
| | Sharing clothes | 4 (1.0) | 1(0.2) | 1(0.2) | |
| | Others | 31 (7.5) | 10(2.4) | 23(5.6) | |
| Condom use is effective ways of preventing STIs | No | 35 (8.4) | 12(2.9) | 23(5.6) | 0.781 |
| | Yes | 289 (69.5) | 90(21.8) | 199(47.8) | |
| | Don't know | 92 (22.1) | 26(6.2) | 66(15.7) | |
| STIs including HIV can transmit through hugging and shaking hands | No | 298 (71.6) | 96(23.2) | 202(48.6) | 0.594 |
| | Yes | 25 (6.0) | 7(1.6) | 18(4.3) | |
| | Don't know | 93(22.4) | 25(6.0) | 68(16.3) | |
| Use of contraceptive pills can reduce risk of STIs | No | 199 (47.8) | 70(16.8) | 129(31.0) | 0.024 |
| | Yes | 53 (12.7) | 20(4.8) | 33(7.9) | |
| | Don't know | 164 (39.4) | 38(9.1) | 126(30.3) | |
| STIs increase the risk of HIV infection | No | 59 (14.2) | 14(3.4) | 45(10.8) | 0.234 |
| | Yes | 239 (57.5) | 81(19.5) | 158(38.0) | |
| | Don't Know | 118 (28.4) | 33(7.9) | 85(20.4) | |
| It safe to have sexual intercourse without using a condom once in a life time | No | 257 (61.8) | 82(19.7) | 175(42.1) | 0.554 |
| | Yes | 71 (17.1) | 18(4.3) | 53(12.7) | |
| | Don't Know | 88 (21.2) | 28(6.7) | 60(14.4) | |
| At higher risk for STI | Male | 94 (22.6) | 36(8.7) | 58(13.9) | 0.062 |
| | Female | 230 (55.3) | 60(14.4) | 170(40.8) | |
| | Don't know | 92 (22.1) | 32(7.8) | 60(14.4) | |
| Alcohol intake increase risk of STIs | No | 74 (17.8) | 19(4.6) | 55(13.2) | 0.547 |
| | Yes | 247(59.4) | 80(19.2) | 167(40.2) | |
| | Don't know | 95 (22.8) | 29(6.9) | 66(15.9) | |

STIs: Sexual transmitted infection, HIV: Human immunodeficiency virus

was significantly associated with factors such as sex (females), not able to identify alcohol intake as a risk factor, and practice of transactional sex.

The overall seroprevalence of STIs detected in the current study (11.5%) is comparable to the study from Jimma (14.3%) which was conducted among clinically suspected University students [22]. In addition, a high prevalence of STIs (18.2%) was reported among students attending Gondar University [23], and India (24%) [24]. The high prevalence of STI among University students in Ethiopia as compared to our study could be due to variation in the level of awareness about the disease and health education on STIs which is provided may not be sufficient or it is overlooked. As University students are far from their parents, they may engage in unsafe sexual activity thinking that they are independent. But it has to be noted some STIs might have been acquired even before they join University and may not reflect student activities at University. Moreover, our finding is high compared to self-reported STIs reported among girls from sub-Saharan Africa (6.92%) [25]

The finding of the current study is higher than a report from India where 5.45%, 6.36%, 1.36%, and 0.45% were reactive for HIV, VDRL, HBV, and HCV respectively [26]. The

**Table 3. Behavior-related characteristics of youth-friendly services Attendees at selected health facilities in Hawassa city, Sidama Regional State, Ethiopia, May 1 to August 30, 2021 (N = 416).**

| Variables | Category | Total N (%) | Sex | | p-value |
|---|---|---|---|---|---|
| | | | Male n(%) | Female n(%) | |
| Tested for HIV | No | 321 (77.2) | 99(23.9) | 222(53.3) | 0.953 |
| | Yes | 95 (22.8) | 29(6.9) | 66(15.9) | |
| Experience of sex without using condom | No | 315 (75.7) | 100(24.0) | 215(51.7) | 0.446 |
| | Yes | 101 (24.3) | 28(6.7) | 73(17.6) | |
| Chew a Khat | No | 397 (95.4) | 114(27.4) | 283(68.0) | 0.000 |
| | Yes | 19 (4.6) | 14(3.4) | 5(1.2) | |
| History of smoking cigarette | No | 414 (99.5) | 127(30.5) | 287(69.0) | 0.555 |
| | Yes | 2 (0.5) | 1(0.2) | 1(0.2) | |
| History of STIs in the last one year | No | 415 (99.8) | 128(31.0) | 287(69.0) | 0.504 |
| | Yes | 1 (.2) | 0(0.0) | 1(0.2) | |
| Engaged in Transactional sex in the last one year | No | 401 (96.4) | 123(29.6) | 280(67.3) | 0.542 |
| | Yes | 15 (3.6) | 5(1.2) | 8(1.9) | |
| Shared sharp materials with others | No | 406 (97.6) | 127(30.5) | 279(67.1) | 0.150 |
| | Yes | 10 (2.4) | 1(0.2) | 9(2.2) | |
| Experience signs and symptoms of STI in the past two weeks | No | 384 (92.3) | 117(28.1) | 267(64.2) | 0.413 |
| | Yes | 32 (7.7) | 11(2.6) | 21(5.1) | |
| History of treatment for STI in the past one year | No | 414 (98.8) | 127(30.5) | 284(68.3) | 0.600 |
| | Yes | 5 (1.2) | 1(0.2) | 4(1.0) | |

difference could be due to the study setting and socio-demographic characteristics. In addition to this, the variation in seroprevalence of HBV and HCV among studies might be due to disparity in the nature of the population studied and the study period [23, 26, 27].

**Table 4. Bivariable and multivariable analysis of factors associated with sero-prevalence of sexual transmitted infection among youth-friendly services Attendees at selected health facilities in Hawassa city, Sidama Regional State, Ethiopia, May 1 to August 30, 2021.**

| Variables | Category | STI n(%) | | COR (95%CI) | p-value | AOR(95%CI) | p-value |
|---|---|---|---|---|---|---|---|
| | | Yes | No | | | | |
| Age | 15–19 | 15 (9.1) | 149 (90.9) | 1 | | 1 | |
| | 20–24 | 33 (13.1) | 219 (86.9) | 1.497 (0.785, 2.852) | 0.220 | 2.002 (0.990, 4.047) | 0.053 |
| Sex | Male | 8 (6.3) | 120 (93.7) | 1 | | 1 | |
| | Female | 40 (13.9) | 248 (86.1) | 2.419 (1.098, 5.330) | 0.028 | 2.989 (1.272, 7.024) | 0.012 |
| Previous place of residence | Rural | 6 (7.4) | 75 (92.6) | 1 | | 1 | |
| | Urban | 42 (12.5) | 293 (87.5) | 1.792 (0.734, 4.373) | 0.200 | 2.031 (0.796, 5.178) | 0.138 |
| Marital Status | Married | 7 (7.8) | 83 (92.2) | 1 | | 1 | |
| | Single | 41 (12.6) | 285 (87.4) | 1.706 (0.738, 3.943) | 0.212 | 2.385 (0.986, 5.771) | 0.054 |
| Multiple sexual partner increase risk of STIs | Yes | 27 (9.2) | 268 (90.8 | 1 | | | |
| | No | 21 (17.4) | 100 (82.6) | 2.084 (1.127, 3.855) | 0.019 | 1.722 (0.854, 3.472) | 0.129 |
| Alcohol intake increase risk of STIs? | Yes | 21 (8.4) | 230 (91.6) | 1 | | 1 | |
| | No | 27 (16.4) | 138 (83.6) | 2.143 (1.167, 3.936) | 0.014 | 2.393 (1.188, 4.817) | 0.015 |
| Engaged in Transactional sex in the last one year | Yes | 5 (33.3) | 10 (66.7) | 4.163 (1.359, 12.748) | 0.013 | 5.527 (1.629, 18.751) | 0.006 |
| | No | 43 (10.7) | 358 (89.3) | 1 | | 1 | |
| Experience of sex without using condom | Yes | 16 (15.8) | 283 (89.8) | 1 | | 1 | |
| | No | 32 (10.2) | 283 (89.8) | 1.665 (0.871, 3.180) | 0.123 | 1.852 (0.909, 3.775) | 0.090 |

STI: Sexually transmitted infection, COR: Crude odd ratio, AOR: Adjusted odd ration

When we look at the specific prevalence of STIs, the most prevalent STIs in the present study was HBsAg (6.1%), followed by HCV (3.1%), HIV (1.2%), and Syphilis (0.7%). This finding is alarming, in particular positive results for HBV, HIV, HCV, and HIV mean a lot as they have an impact over the course of time, and also the infection can be transmitted to the family and community at large. Unmanaged chronic infection of HBV and HCV can cause hepatocellular carcinoma and cirrhosis. Likewise, uncontrolled HIV infection can lead to depletion of the immune system and predispose to other opportunistic infections.

Our findings disagrees with the prevalence reported from India where the highest prevalence was HIV (5.45%) followed by Syphilis (6.36%), HBV (1.36%) and HCV (0.45%) [27] and report from Dessie, Ethiopia: HIV (7.5%), HBV(13.1%), HCV(0.6%), and Syphilis (12.5%) [26]. A study conducted in a Gondar indicated a high prevalence of HBV (14.6%) and HCV (12.41%) as compared to ours [28]. The possible explanations for the differences are the laboratory methods used and the nature of the population studied [26–28].

The prevalence of Syphilis (0.7%) is low compared to a study done in Brazil (10.09%) [29]. The prevalence of HBsAg (6.1%) is lower than the prevalence reported from the Central African Republic (42.3%) [30]

Among the independent variables analyzed, female participants were about three times more likely to be positive for STIs as compared with their male counterparts (p = 0.012). A similar finding was reported in Nigeria [31]. This could be due to anatomical and physiological differences.

In this study, we have assessed the role of knowledge on STIs; however, only one factor was significantly associated with the occurrence of STIs. Participants who are unable to identify alcohol intake as a risk factor for STIs were about two times more likely to acquire STIs (p = 0.015) which is comparable with findings reported from Iran [32]. A study conducted among students in Southern parts of Ethiopia [33] and Tanzania [34] reported that the study participants who are not aware that alcohol intake is a risk factor for STIs were more likely to acquire STIs.

In this study, we have also assessed the role behaviors/practices on STIs out of which only taking part in transactional sex was significantly associated. Taking part in transactional sex increases the odds of acquiring STIs by about five times. More than 30% of participants who are engaged in transactional sex were positive for STIs as compared to those who did not practice transactional sex (10.7%) (p = 0.006). Similar to the current study, a high prevalence of STIs was reported among participants practicing transactional sex in Liberia [35] and Nigeria [36]. In developing countries, female youth (in most cases) might be forced to engage in transactional sex to solve their economic problem. Most youths prefer to have sex in exchange for money, food, and shelter which makes them susceptible to STIs. Since this group of the population belongs to low economic status and has low awareness about STIs, they may not use protection methods like condom use during sexual intercourse.

Albeit the absence of a significant association between STIs and other factors, the high prevalence of STIs (15.8%) was identified among participants who said that they have never used a condom during sexual intercourse; the prevalence among those who used condoms during the sexual intercourse was (10.2%) (p = 0.09). Not using condoms during sexual intercourse will increase the risk of STIs. A low prevalence of STIs (9.2%) among participants who responded that having many sexual partners is a risk factor for SITs while a high prevalence of STIs among partners who said having many sexual partners will not increase the risk of STIs (17.4%) (p = 0.129) was also observed. Having multiple sexual partners will increase the odds of STIs.

In the present study, 8.4% and 17.1% of participants think that the use of condoms is not effective in preventing STIs and it is safe to have sex without using condoms respectively. On

top of that, 24.3% of participants responded that they have sex without using a condom. 12.7% of participants responded that the use of contraceptives will reduce STIs. All these indicate that the knowledge participants have regarding STIs is questionable and should be emphasized by all stakeholders.

## Policy and program recommendations

Based on the fining of the current study the following recommendations were made

- Find the rout cause of high STIs among youths in Hawassa city and act accordingly.

- Provide health education (awareness) to youths about STIs: transmission, prevention, and damage they cause

## Limitations of the study

As the participants of this study are facility-attending youth it may not represent other youths who were not attending the facility. A face-to-face interview might have biased responses from participants. As some variables ask about the history, there might be recall bias. We have defined transactional as heterosexual intercourse only which is inclusive as there are different forms of sexual intercourse. Infections included in STIs operational definition is not exhaustive, moreover, there are other ways of transmission for STIs other than sexual contacts such as blood transfusion and needle injury.

## Conclusions

In the present study, the overall prevalence of sexually transmitted infections among youth attending friendly services at selected health facilities in Hawassa city was high. The majority of participants were positive for HBsAg. Policymakers should find the reason for the high prevalence of HBsAg and consider prevention strategies which may include vaccination and health education. Being female, unable to identify Alcohol intake as a risk for STIs, and practice of transactional sex were significant predictors for STIs. High numbers of participants think that the use of condoms is not effective in preventing STIs. As this is very concerning, we recommend further large-scale study is needed to determine knowledge, attitude, and practice on STIs prevention methods. Based on the finding of this study, we recommend further study to identify the route cause for high burden of STIs among study group and provision of health education about STI which includes its transmission, prevention and damage they may cause.

## Supporting information

**S1 Data.**
(SAV)

## Acknowledgments

We thank all staff of the participating sites for their assistance during data collection. We also thank all study participants for their willingness to take part in this study.

## Author Contributions

**Conceptualization:** Erdachew Ambaye, Moges Desta Ormago, Musa Mohammed Ali.

Formal analysis: Erdachew Ambaye.

Investigation: Erdachew Ambaye, Musa Mohammed Ali.

Methodology: Erdachew Ambaye, Musa Mohammed Ali.

Project administration: Erdachew Ambaye.

Resources: Erdachew Ambaye.

Supervision: Moges Desta Ormago, Musa Mohammed Ali.

Writing – original draft: Musa Mohammed Ali.

Writing – review & editing: Moges Desta Ormago, Musa Mohammed Ali.

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
