## [Decision Letter · Decision Letter 0]

26 Sep 2022

PONE-D-22-12316Sero-prevalence and associated factors of sexually transmitted infections among youth-friendly services attendantsPLOS ONE

Dear Dr. Ali,

Thank you for submitting your manuscript to PLOS ONE. After careful consideration, we feel that it has merit but does not fully meet PLOS ONE’s publication criteria as it currently stands. Therefore, we invite you to submit a revised version of the manuscript that addresses the points raised during the review process.

 Your manuscript has been assessed by two peer-reviewers and their comments are appended below.  The reviewers comment that your manuscript requires further information and/or clarification on the methodology and statistics used in this study. In addition, the reviewers have raised major concerns with the presentation of the results and discussion of this study. Furthermore, the reviewers have commented that your manuscript does not adequately discuss the limitations to this study.  Could you please revise the manuscript to carefully address the concerns raised?

We look forward to receiving your revised manuscript.

Kind regards,

Maria Elisabeth Johanna Zalm, Ph.D

Editorial Office

PLOS ONE

2.Thank you for stating the following in your Competing Interests section: 

“no competing interests exist.”

3.In your Data Availability statement, you have not specified where the minimal data set underlying the results described in your manuscript can be found. PLOS defines a study's minimal data set as the underlying data used to reach the conclusions drawn in the manuscript and any additional data required to replicate the reported study findings in their entirety. All PLOS journals require that the minimal data set be made fully available. For more information about our data policy, please see http://journals.plos.org/plosone/s/data-availability.

4.We note that you have indicated that data from this study are available upon request. PLOS only allows data to be available upon request if there are legal or ethical restrictions on sharing data publicly. For more information on unacceptable data access restrictions, please see http://journals.plos.org/plosone/s/data-availability#loc-unacceptable-data-access-restrictions.

5.Please upload a copy of Figure 4, to which you refer in your text on page 15. If the figure is no longer to be included as part of the submission please remove all reference to it within the text.

Reviewers' comments:

Reviewer's Responses to Questions

**Comments to the Author**

1. Is the manuscript technically sound, and do the data support the conclusions?

Reviewer #1: Yes

Reviewer #2: Partly

2. Has the statistical analysis been performed appropriately and rigorously? 

Reviewer #1: Yes

Reviewer #2: No

3. Have the authors made all data underlying the findings in their manuscript fully available?

Reviewer #1: Yes

Reviewer #2: No

4. Is the manuscript presented in an intelligible fashion and written in standard English?

Reviewer #1: Yes

Reviewer #2: No

5. Review Comments to the Author

Reviewer #1: The paper is generally well written and structured. However, in my opinion the paper has some shortcomings in regards to some data analyses and text, and I feel this unique dataset has not been utilized to its full extent. Below I have provided numerous remarks on the text as it is often vague and long-winded. In the abstract please state your recommendation.

1) How can study only 4 microorganisms (HIV, HBV, HCV and T.palladium) represent STI.

2) Why you were use 50% p-value to calculate the sample size? Because there are a lot of study conducted in your country.

3) Study about N.gonorrhoeae is you objective? If it is your objective the result obtained must be included into total number of STI. Unless it better to delete.

4) Please edit your operational definition for STI as “STI is defined as a positive status either for single or combination of infections specific to HIV, HBV, HCV, and syphilis”.

5) Laboratory test methods you were used are not Gold standard except for the culture. Therefore, what types of quality assurance were conducted to believe your finding?

6) Some Variables in table were not clear and not specific (history of STIs, Recent signs and symptoms of STI). Please show the exact period.

7) Include P-value of COR in the table 3.

8) To state you result for syphilis to was prefer to use the word “RPR’. What is the reason?

9) On page 15 line 267 you cite Figure 4. However, there is no any figure in the manuscript.

10) In Discussion part: page 18 line 288-291. According to you result the CI for the prevalence of STI was 8.7%- 14.9%. Therefore, how can it be lower than study conducted from Jimma (14.3%).

11) Discussion is good. But it very nice if the sequence is similar with subheading of result ( Socio-demographic, knowledge and behavior, sero-prevalence, risk factor)

12) In discussion part, most of your justification for your finding does not have references.

13) Authors’ contribution: from three authors only two of them were approve this manuscript. Why not all.

14) instances I also suggested to cite more relevant and recent literature

15) Correct your reference style

Reviewer #2: This an excellent start to an important article that highlights the high STI burden among youth in Ethiopia.

I do have some major concerns, especially pertaining to the presentation of results and discussion.

In general:

Be careful with sweeping statements like at-risk partners (what does this mean?). There are other examples throughout.

- the manuscript should be checked for grammar, writing and capitalization of proper nouns.

- The inclusion of swabbing urogenital sites (which ones?) was included briefly in methods and results write up, but does not seem pertinent to seroprevalence and not discussed in the abstract or discussion. Suggest taking this section out or including it fully.

- all acronyms should be defined at first use

General Ethical issues:

- I strongly suggest not mentioning the names of the sites that were included. This could lead to possible identification of participants, and on a worldwide scale, these names are not important.

- How were consents from minor participants given? To a guardian? What was the process?

Title: you briefly present gonorrhoeae findings… either take them out or include because they are not part of seroprevalence.

Background:

The introduction/background section should be reorganized:

- There is quite a bit of information that is not pertinent at this level of manuscript. Focus on: presenting the problem and solutions, the nature and scope of the problem investigated, a pertinent literature review to orient the reader, and the aims/objectives (https://guides.lib.uci.edu/c.php?g=334338&p=2249903). In general- why are adolescents and young adults, especially females at increased risk for STI acquisition (and which types of STIs—the epidemiology of each varies so widely that they should not be grouped like this), what is known in Ethiopia already and why is it important to look into adolescents as well, how do you plan to study this population and these infections (last part of last paragraph of the intro).

- Each paragraph should focus on one topic (example: paragraph 3 jumps from youth and stis to economic burden). Stick to one topic per paragraph.

- what are ´list income countries? (line 86)

Methods

Participant inclusion should be better discussed. How many were approached, how many declined? How did you end up with the final sample. Consider a flowchart to show this.

The sample collection and lab procedures vary between too much detail and not enough. I.e. you don´t need to go into the details of how to read a immunochromatic test strip but antimicrobial susceptibility testing should be better described.

- Why was RPR only used? And not (as WHO suggests) a RPR then antibody test (rapid test)?

- What dilutions were carried out for RPR among all participants?

-Line 170 describes yet again the swabs? Only describe in one part of this section.

- Line 171 ´urogenital infection´ can mean different things to different people. What were the criteria and who made the call as to whether to include this testing or not?

- Line 188 who did consistency testing

-Line 189 are ´trained health professionals´ part of the author list? Indicate whom please

-Line 217 and 218: cut point for what?

- Check if operational definitions should be in this manuscript…if so, this seems like a very incomplete list. Please use a variable list that is complete and describes the measures correctly. Why only include heterosexual intercourse as transactional sex--This seems biased?

The variable list in the tables of the results is vague and needs better explanation.

Results:

- age should be presented as a medium with IQR, (not mean with SD)

- please do not repeat results from the tables in writeup of results.

- Tables: Present tables 1 and 2 segregated by sex with p-values to show differences in demographics and behaviours.

-Group tables by knowledge/beliefs and behaivours. As Table 2 is now, it seems to jump around and is very hard to follow.

- there needs to be a sex-segregated table for STI prevalence with p-values.

- please report RPR findings by titer.

- Please report co-infections of STI- this is more important even than the % of STI by

- Sti prevalence should be presented first by STI then as grouped STI (not grouped STI first)

- Please use proper description of odds ratios: https://www.ncbi.nlm.nih.gov/pmc/articles/PMC2938757/ OR are not ´time higher than´ but rather : OR=1 Exposure does not affect odds of outcome, OR>1 Exposure associated with higher odds of outcome, OR<1 Exposure associated with lower odds of outcome´

Discussion:

- This section should be reorganized. https://www.jclinepi.com/article/S0895-4356(13)00193-5/fulltext please follow in order to have a better organization of each paragraph. Again, please only include one topic per paragraph after paragraph 1. Example: Paragraph 1 should be an overview of major findings and results, etc.

- Comparing chlamydia, gonorrhea and RPR findings to other studies that use distinct methods may not be accurate. Please check if PCR was not used for CT/NG on papers you cite, as well as rapid testing+RPR results. Make sure methods are similar.

- try not to compare with countries and studies in other regions that may not be comparable (i.e. focus on low and middle income countries- there are some published papers that are similar in Latin America- you can compare from there more accurately than the US.

- behaviour outcomes and sti associations need to be discussion separately from knowledge/belief outcomes and sti associations as these phenomena and interventions are different from behavioural phenomena and interventions.

-There needs to be a clear policy and programme section to indicate how to better the sti outcomes in this population. I.e. HBsAg was very high—let´s focus on vaccination and followup studies on vaccination status to understand what populations are not being vaccinated.

Limitations:

There are quite a few limitations to this study, which were not at all included in a section at the end of the discussion. For example:

- you´ve included facility-attending youth. These are not any youth, and they most likely have received greater sexuality education or have greater access to healthcare in the past. This needs to be discussed.

- interview face to face questionnaire biases responses of participants. Please discuss how.

- discuss all other limitations fully.

6. PLOS authors have the option to publish the peer review history of their article (what does this mean?). If published, this will include your full peer review and any attached files.

Reviewer #1: No

Reviewer #2: No

---

## [Author Response · Author response to Decision Letter 0]

5 Oct 2022

Could you please revise the manuscript to carefully address the concerns raised?

Response: We have included the limitation suggested.

We look forward to receiving your revised manuscript.

Kind regards,

Maria Elisabeth Johanna Zalm, Ph.D

Editorial Office

PLOS ONE

Response: OK

2.Thank you for stating the following in your Competing Interests section: 

“no competing interests exist.”

Response: We have included as suggested. 

3.In your Data Availability statement, you have not specified where the minimal data set underlying the results described in your manuscript can be found. PLOS defines a study's minimal data set as the underlying data used to reach the conclusions drawn in the manuscript and any additional data required to replicate the reported study findings in their entirety. All PLOS journals require that the minimal data set be made fully available. For more information about our data policy, please see http://journals.plos.org/plosone/s/data-availability.

Response: We have uploaded raw data used for analysis. 

4.We note that you have indicated that data from this study are available upon request. PLOS only allows data to be available upon request if there are legal or ethical restrictions on sharing data publicly. For more information on unacceptable data access restrictions, please see http://journals.plos.org/plosone/s/data-availability#loc-unacceptable-data-access-restrictions.

Response: We have uploaded raw data used for analysis. 

5.Please upload a copy of Figure 4, to which you refer in your text on page 15. If the figure is no longer to be included as part of the submission please remove all reference to it within the text.

Reviewers' comments:

Reviewer's Responses to Questions

Comments to the Author

1. Is the manuscript technically sound, and do the data support the conclusions?

Reviewer #1: Yes

Reviewer #2: Partly

2. Has the statistical analysis been performed appropriately and rigorously?

Reviewer #1: Yes

Reviewer #2: No

3. Have the authors made all data underlying the findings in their manuscript fully available?

Reviewer #1: Yes

Reviewer #2: No

4. Is the manuscript presented in an intelligible fashion and written in standard English?

Reviewer #1: Yes

Reviewer #2: No

5. Review Comments to the Author

Reviewer #1: The paper is generally well written and structured. However, in my opinion the paper has some shortcomings in regards to some data analyses and text, and I feel this unique dataset has not been utilized to its full extent. Below I have provided numerous remarks on the text as it is often vague and long-winded. In the abstract please state your recommendation.

1) How can study only 4 microorganisms (HIV, HBV, HCV and T.palladium) represent STI.

Response: Thank you for the comment. We did not mean HIV, HBV, HCV, and T. palladium represent STI. We operationally defined STI for this study. Moreover we have mentioned the limitation of operational definition. 

2) Why you were use 50% p-value to calculate the sample size? Because there are a lot of study conducted in your country.

Response: Yes it is true there several studies which separately addressed STI in non-adolescent population and there are few studies of this kind among the study group. As we have included >2 disease to describe STI we have used a prevalence 50% to represent all of them as it will give maximum sample size. 

3) Study about N. gonorrhoeae is you objective? If it is your objective the result obtained must be included into total number of STI. Unless it better to delete.

Response: We accept the comment; accordingly have deleted information related to N. gonorrhoeae

4) Please edit your operational definition for STI as “STI is defined as a positive status either for single or combination of infections specific to HIV, HBV, HCV, and syphilis”.

Response: We have corrected accordingly as shown in the manuscript with track change. 

5) Laboratory test methods you were used are not Gold standard except for the culture. Therefore, what types of quality assurance were conducted to believe your finding?

Response: in addition to inbuilt quality control, we used External quality control sample from blood bank and public health institute laboratory to verify test kits. 

6) Some Variables in table were not clear and not specific (history of STIs, Recent signs and symptoms of STI). Please show the exact period.

Response: We have corrected accordingly for both variables.

7) Include P-value of COR in the table 3.

Response: we have included it. 

8) To state you result for syphilis to was prefer to use the word “RPR’. What is the reason?

Response: We used ‘RPR’ instead of ‘syphilis’ because being reactive for RPR may not indicate active disease ‘syphilis’. To reduce confusion we replaced ‘RPR’ with ‘syphilis’ 

9) On page 15 line 267 you cite Figure 4. However, there is no any figure in the manuscript.

Response: We have removed figure 4 as we have not used it. 

10) In Discussion part: page 18 line 288-291. According to you result the CI for the prevalence of STI was 8.7%- 14.9%. Therefore, how can it be lower than study conducted from Jimma (14.3%).

Response: We agree with the comment and corrected it as shown in the manuscript with track change. 

11) Discussion is good. But it very nice if the sequence is similar with subheading of result ( Socio-demographic, knowledge and behavior, sero-prevalence, risk factor)

Response: Thank you for the comment; we have organized the discussion section based on our objectives: seroprevalence of STI and Factors associated with it. We believe socio-demographic and behavioural variables are addressed in the risk factor analysis. 

12) In discussion part, most of your justification for your finding does not have references.

Response: We have compared the finding of the current study with other studies. We tried to include references wherever they are relevant. We have included our opinion after assessing ours and others that is why some description lacks references. 

13) Authors’ contribution: from three authors only two of them were approve this manuscript. Why not all.

Response: Thank you for the comment, we have corrected as suggested. 

14) instances I also suggested to cite more relevant and recent literature

Response: There are scarce similar published articles from Ethiopia. We have tried to include relevant publication as shown in the manuscript with track change. We have added references in the revised manuscript. 

15) Correct your reference style

Response: we have revised the reference style

Reviewer #2: This an excellent start to an important article that highlights the high STI burden among youth in Ethiopia.

I do have some major concerns, especially pertaining to the presentation of results and discussion.

In general:

Be careful with sweeping statements like at-risk partners (what does this mean?). There are other examples throughout.

Response: Thank you for the comment we have tried to remove or correct those phrases as shown in the manuscript with track change. 

- the manuscript should be checked for grammar, writing and capitalization of proper nouns.

Response: We revised the English as shown in the manuscript with track change. 

- The inclusion of swabbing urogenital sites (which ones?) was included briefly in methods and results write up, but does not seem pertinent to seroprevalence and not discussed in the abstract or discussion. Suggest taking this section out or including it fully.

Response: As suggested by both reviewers we have removed information related to N. gonorrhoeae

- all acronyms should be defined at first use

Response: We have revised as suggested. 

General Ethical issues:

- I strongly suggest not mentioning the names of the sites that were included. This could lead to possible identification of participants, and on a worldwide scale, these names are not important.

Response: We accept the comment; we have omitted the name of the study sites. 

- How were consents from minor participants given? To a guardian? What was the process?

Response: as we have indicated in the manuscript, for minors, both consent from their parents and assent from minors was obtained before data collection. We have invited parents of minors to the health facility to provide consent for their children. 

Title: you briefly present gonorrhoeae findings… either take them out or include because they are not part of seroprevalence.

Response: as we have mentioned above we removed information related to gonorrhoeae from the manuscript. 

Background:

The introduction/background section should be reorganized:

Response: We revised as suggested 

- There is quite a bit of information that is not pertinent at this level of manuscript. Focus on: presenting the problem and solutions, the nature and scope of the problem investigated, a pertinent literature review to orient the reader, and the aims/objectives (https://guides.lib.uci.edu/c.php?g=334338&p=2249903) 

Response: Baaed on the comment we have revised the background

 In general- why are adolescents and young adults, especially females at increased risk for STI acquisition (and which types of STIs—the epidemiology of each varies so widely that they should not be grouped like this), what is known in Ethiopia already and why is it important to look into adolescents as well, how do you plan to study this population and these infections (last part of last paragraph of the intro).

Response: Baaed on the comment we have revised the back ground

- Each paragraph should focus on one topic (example: paragraph 3 jumps from youth and stis to economic burden). Stick to one topic per paragraph.

Response: We accept the comment, we broke paragraph 3 in two and we tried to link STI among youth and economic burden as shown in the manuscript with track change. 

- what are ´list income countries? (line 86)

Response: we accept the comment we corrected as ‘low income countries’ 

Methods

Participant inclusion should be better discussed. How many were approached, how many declined? How did you end up with the final sample. Consider a flowchart to show this.

Response: the excepted number of source population during the study period was 2102. We proportionally allocated the sample size to all (four) study sites. We used systematic random sampling technique to recruit study participants. Whenever the client was not willing to participate we take the next person. We have used single proportion formula to calculate sample size. In case of refusal we go for the next client. 

The sample collection and lab procedures vary between too much detail and not enough. I.e. you don´t need to go into the details of how to read a immunochromatic test strip but antimicrobial susceptibility testing should be better described.

Response: We accept the comment; we have omitted the details of the lab. We have omitted information related to Gonorrhoea. 

- Why was RPR only used? And not (as WHO suggests) a RPR then antibody test (rapid test)?

Response: We did both as we have mentioned in the manuscript with a track change. 

- What dilutions were carried out for RPR among all participants?

Response: We did not perform dilution 

-Line 170 describes yet again the swabs? Only describe in one part of this section.

Response: we have omitted information related to NG

- Line 171 ´urogenital infection´ can mean different things to different people. What were the criteria and who made the call as to whether to include this testing or not?

Response: we have omitted information related to NG

- Line 188 who did consistency testing

Response: the consistency was checked by the principal investigator 

-Line 189 are ´trained health professionals´ part of the author list? Indicate whom please

Response: No, they were not part of the authors list they are recruited for data collection. 

-Line 217 and 218: cut point for what?

Response: it is a cut point for determination of statistically significant association as modified in the manuscript with track change. 

- Check if operational definitions should be in this manuscript…if so, this seems like a very incomplete list. Please use a variable list that is complete and describes the measures correctly. Why only include heterosexual intercourse as transactional sex--This seems biased?

The variable list in the tables of the results is vague and needs better explanation.

Response: we accept the comment we included in the limitation section, we have also revised naming of variables listed in tables. 

Result: 

Results:

- age should be presented as a medium with IQR, (not mean with SD)

Response: we have corrected it. 

- please do not repeat results from the tables in writeup of results.

Response: As recommended we have omitted some information presented in the text. 

- Tables: Present tables 1 and 2 segregated by sex with p-values to show differences in demographics and behaviours.

Response: we have segregated table 2 and 3 (previous table 2 now it becomes two tables as you suggested). 

-Group tables by knowledge/beliefs and behaivours. As Table 2 is now, it seems to jump around and is very hard to follow.

Response: based on the comment we have segregated them as table 2 for knowledge and table 3 fro behaviours and others.

- there needs to be a sex-segregated table for STI prevalence with p-values.

Response: in previous table 3 (now it is table four) in row 3 we have indicated distribution of prevalence of STI among males (6.3%) and females (13.9%). If we segregate other variables by sex it will complicate the table and also it is not the objective of our study. 

- please report RPR findings by titer.

Response: we did not perform RPR titer 

- Please report co-infections of STI- this is more important even than the % of STI by

Response: we have included confection under heading seroprevalence of STI

- Sti prevalence should be presented first by STI then as grouped STI (not grouped STI first)

Response: we have corrected it accordingly. 

- Please use proper description of odds ratios: https://www.ncbi.nlm.nih.gov/pmc/articles/PMC2938757/ OR are not ´time higher than´ but rather : OR=1 Exposure does not affect odds of outcome, OR>1 Exposure associated with higher odds of outcome, OR<1 Exposure associated with lower odds of outcome´

Response: We have corrected it accordingly as shown in the manuscript with track change. 

Discussion:

- This section should be reorganized. https://www.jclinepi.com/article/S0895-4356(13)00193-5/fulltext please follow in order to have a better organization of each paragraph. Again, please only include one topic per paragraph after paragraph 1. Example: Paragraph 1 should be an overview of major findings and results, etc.

Response: We have revised the discussion as shown in the manuscript with track change. 

- Comparing chlamydia, gonorrhea and RPR findings to other studies that use distinct methods may not be accurate. Please check if PCR was not used for CT/NG on papers you cite, as well as rapid testing+RPR results. Make sure methods are similar.

Response: We accept the comment we have removed a study on gonorrhoea and chlamydia. Wherever it is applicable we have mentioned difference in laboratory methods for discrepancy. 

- try not to compare with countries and studies in other regions that may not be comparable (i.e. focus on low and middle income countries- there are some published papers that are similar in Latin America- you can compare from there more accurately than the US.

Response: We accept the comment we omitted information from USA.

- behaviour outcomes and sti associations need to be discussion separately from knowledge/belief outcomes and sti associations as these phenomena and interventions are different from behavioural phenomena and interventions.

Response: In the study we have identified 3 factors which are significantly associated. We have tried to segregate them as suggest. 

-There needs to be a clear policy and programme section to indicate how to better the sti outcomes in this population. I.e. HBsAg was very high—let´s focus on vaccination and followup studies on vaccination status to understand what populations are not being vaccinated.

Response: we agree with the comment accordingly we have include ‘Policy maker should find the reason for high prevalence of HBsAg and consider prevention strategies which may include vaccination and health education’ 

Limitations:

There are quite a few limitations to this study, which were not at all included in a section at the end of the discussion. For example:

- you´ve included facility-attending youth. These are not any youth, and they most likely have received greater sexuality education or have greater access to healthcare in the past. This needs to be discussed.

- interview face to face questionnaire biases responses of participants. Please discuss how.

- discuss all other limitations fully.

Response: We have included the limitations of the study after the discussion section.

---

## [Decision Letter · Decision Letter 1]

2 Nov 2022

PONE-D-22-12316R1Sero-prevalence and associated factors of sexually transmitted infections among youth-friendly services attendantsPLOS ONE

Dear Dr. Ali,

Thank you for submitting your manuscript to PLOS ONE. After careful consideration, we feel that it has merit but does not fully meet PLOS ONE’s publication criteria as it currently stands. Therefore, we invite you to submit a revised version of the manuscript that addresses the points raised during the review process.

We look forward to receiving your revised manuscript.

Kind regards,

Chuanyi Ning, Ph.D.

Academic Editor

PLOS ONE

Reviewers' comments:

Reviewer's Responses to Questions

**Comments to the Author**

1. If the authors have adequately addressed your comments raised in a previous round of review and you feel that this manuscript is now acceptable for publication, you may indicate that here to bypass the “Comments to the Author” section, enter your conflict of interest statement in the “Confidential to Editor” section, and submit your "Accept" recommendation.

Reviewer #1: All comments have been addressed

Reviewer #2: All comments have been addressed

2. Is the manuscript technically sound, and do the data support the conclusions?

Reviewer #1: Yes

Reviewer #2: Partly

3. Has the statistical analysis been performed appropriately and rigorously? 

Reviewer #1: Yes

Reviewer #2: Yes

4. Have the authors made all data underlying the findings in their manuscript fully available?

Reviewer #1: Yes

Reviewer #2: No

5. Is the manuscript presented in an intelligible fashion and written in standard English?

Reviewer #1: Yes

Reviewer #2: No

6. Review Comments to the Author

Reviewer #1: Now your manuscript is very good.

The researchers have been addressed all comments accordingly.

Reviewer #2: In general, this is better written and described than the first version. However the authors should be careful of English use, especially odd capitalizations throughout.

It may be very helpful to find other studies published with Plos that are similar to see how the paragraph, especially the discussion is organized.

Title Attendees vs attendants- check definitions

The section on knowledge and beliefs is big in the results section of the paper, but the title does not mention it- perhaps consider including.

Abstract Please check English throughout.

Objective: to describe the seroprevalence of (which STIs)

Methods: do not capitalize syphilis, hepatitis B or hepatitis C. Define all acronyms you use in the abstract when you first mention them

Results: please give individual STI results first, then grouped.

Consider using confidence interval vs confidence level

Give % and in (n/N). Don´t start a sentence with a number. What does the female AOR mean? Only use OR to two spots after the decimal. What is adjustment for? The section on OR needs to be rewritten as previously suggested to correctly report OR and AOR.

Conclusions: comparable to other studies where and with what population?

You still need to rewrite association results more clearly. STIs were associated with x not x with STI. Use terms more descriptive than ´High numbers´--- and there is an ´association with sexual activity without cndoms? What does the last line mean?

Background You describe the effects of STIs that you do not include (i.e.ectopic pregnancy, infertility, pelvic pain, inflammatory). Focus this on what you are studying only.

Syphilis _may_ cause but doesn´t always or even usually cause neurological, cardiovascular or derm disease, same with neonatal effects).

Broad statements such as STI prevalence ranges from 9-21% don’t say a whole lot. Focus on the STIs youstudy only, not all STIs in broad statements.

Consider low and middle income country vs developing country.

The STIs you study don´t significantly affect reproductive potential (line 87).

The scientific community prefers to use risk reduction rather than abstinence focuse (line 91-93)

Please give more background on Ethiopia and Hawassa in the introduction (not in methods)

Methods Please give types of HIV tests (antibody vs antigen? Lines 151+

Data analysis: what % of data were rechecked for correctness?

only define acronyms at first use from introduction onwards.

Consent: In minors: who gave consent before data collection? Minors or guardians?

Results Did all participants answer all qustions? If not, n/N needs to be given for each variable.

% can be given before n (and put n in paranthases)

For the tables: consider not presenting in question format, but instead as reported HIV test, Has chewed khat, has smoked cigarettes, etc.

Table 1: the variable monthly income is household or individual?

Table 2: what does the p-value indicate? Difference between sexes? Include in the table title what it means.

Again, make sure the acronyms for STIs have been presented previously.

Factors associated with STIs: no need to reexplain methods.

´Grouped STs were associated with x, y and z or x, y and z were associated with increased odds of grouped STI – and make sure you´ve defined what grouped STIs are in methods)

Discussion Have a look again at what to include in the first paragraph of a discussion- you can include some overall riskfactors as well as STI results.

Lines starting 250: are these studies with reports of STI? Or findings in laboratory testing? It seems like as It is written that participants of these studies had to self-report their sti diagnosis (which is never going to be accurate)

Paragraph 3 needs to be restructured. Keep to one topic. The highest prevalence is HBV- what does that mean for vaccination? Further research should look at vaccination status.

All the paragraphs with comparisons from other LMIC can be grouped into one paragraph.

Make sure your study is comparable to what you present. What is given from Rhode Island doesn´t make sense (in incarcerated populations)

Make sure your comparisons make sense. Your syphilis prevalence of 0.7% does to compare with Brazilian prevalence.

You should have a section on ´policy and program recommendations´ or at the very least be mentioning this throughout the discussion. A whole section would be better though.

Limitations: use proper limitations and bias vocabulary https://jech.bmj.com/content/58/8/635

Conclusions: you need to include program recommendations.

Please define abbreviations within text as well (when you first use them)

7. PLOS authors have the option to publish the peer review history of their article (what does this mean?). If published, this will include your full peer review and any attached files.

Reviewer #1: No

Reviewer #2: No

---

## [Author Response · Author response to Decision Letter 1]

3 Nov 2022

PONE-D-22-12316R1

Sero-prevalence and associated factors of sexually transmitted infections among youth-friendly services attendants

PLOS ONE

Dear Dr. Ali,

Thank you for submitting your manuscript to PLOS ONE. After careful consideration, we feel that it has merit but does not fully meet PLOS ONE’s publication criteria as it currently stands. Therefore, we invite you to submit a revised version of the manuscript that addresses the points raised during the review process.

We look forward to receiving your revised manuscript.

Kind regards,

Chuanyi Ning, Ph.D.

Academic Editor

PLOS ONE

Reviewers' comments:

Reviewer's Responses to Questions

Comments to the Author

1. If the authors have adequately addressed your comments raised in a previous round of review and you feel that this manuscript is now acceptable for publication, you may indicate that here to bypass the “Comments to the Author” section, enter your conflict of interest statement in the “Confidential to Editor” section, and submit your "Accept" recommendation.

Reviewer #1: All comments have been addressed

Reviewer #2: All comments have been addressed

2. Is the manuscript technically sound, and do the data support the conclusions?

Reviewer #1: Yes

Reviewer #2: Partly

3. Has the statistical analysis been performed appropriately and rigorously?

Reviewer #1: Yes

Reviewer #2: Yes

4. Have the authors made all data underlying the findings in their manuscript fully available?

Reviewer #1: Yes

Reviewer #2: No

Response: we have uploaded the raw data in SPSS

5. Is the manuscript presented in an intelligible fashion and written in standard English?

Reviewer #1: Yes

Reviewer #2: No

6. Review Comments to the Author

Reviewer #1: Now your manuscript is very good.

The researchers have been addressed all comments accordingly.

Response: Thank you for acknowledging our revised manuscript 

Reviewer #2: In general, this is better written and described than the first version. However the authors should be careful of English use, especially odd capitalizations throughout.

It may be very helpful to find other studies published with Plos that are similar to see how the paragraph, especially the discussion is organized.

Response: Thank you for giving us additional comments. We have consulted other papers and corrected accordingly as shown in manuscript with track change. 

Title Attendees vs attendants- check definitions

Response: Thank you for the comment, we have checked the definition we found ‘Attendees’ appropriate. 

The section on knowledge and beliefs is big in the results section of the paper, but the title does not mention it- perhaps consider including.

Response: We feel this idea is included under associated factors; moreover if we include this information in the title the title will become lengthy. 

Abstract Please check English throughout.

Response: We have revised. 

Objective: to describe the seroprevalence of (which STIs)

Response: We have corrected as suggested.

Methods: do not capitalize syphilis, hepatitis B or hepatitis C. Define all acronyms you use in the abstract when you first mention them.

Response: We have corrected the capitalization and we defined the acronyms except the name of kits. 

Results: please give individual STI results first, then grouped.

Response: We have presented like this based on based on the previous comment we also would like if stands as it is. Previous comment “Sti prevalence should be presented first by STI then as grouped STI (not grouped STI first)”

Consider using confidence interval vs confidence level

Response: We have used confidence interval (CI)

Give % and in (n/N). Don´t start a sentence with a number. What does the female AOR mean? Only use OR to two spots after the decimal. What is adjustment for? The section on OR needs to be rewritten as previously suggested to correctly report OR and AOR.

Response: We have corrected as suggested. The adjustment was for confounders. 

Conclusions: comparable to other studies where and with what population?

Response: We have modified as shown in the manuscript with track change. 

You still need to rewrite association results more clearly. STIs were associated with x not x with STI. Use terms more descriptive than ´High numbers´--- and there is an ´association with sexual activity without cndoms? What does the last line mean?

Response: We have corrected as suggested. 

Background You describe the effects of STIs that you do not include (i.e.ectopic pregnancy, infertility, pelvic pain, inflammatory). Focus this on what you are studying only.

Response: We have omitted the information as recommended

Syphilis _may_ cause but doesn´t always or even usually cause neurological, cardiovascular or derm disease, same with neonatal effects).

Response: Yes we agree with the comment. We modified as ‘syphilis may cause…’

Broad statements such as STI prevalence ranges from 9-21% don’t say a whole lot. Focus on the STIs youstudy only, not all STIs in broad statements.

Response: We have omitted the statement as recommended. 

Consider low and middle income country vs developing country.

Response: Thank you for the comment. We think we have provided sufficient background for the current study based on the previous comment. 

The STIs you study don´t significantly affect reproductive potential (line 87).

Response: This is not our study; the information was taken from reference # 19 as a background. From our study syphilis may affect reproductive potential. 

The scientific community prefers to use risk reduction rather than abstinence focuse (line 91-93)

Response: We think that starting sexual intercourse at early age without enough knowledge, attitude, and practice about STIs is also a risk factor especially in developing countries which should deserve a due attention. 

Please give more background on Ethiopia and Hawassa in the introduction (not in methods)

Response: We moved some information from the method to background section as suggested. 

Methods Please give types of HIV tests (antibody vs antigen? Lines 151+

Response: We have corrected accordingly. 

Data analysis: what % of data were rechecked for correctness?

Response: 5% (if the question refers to pre-test). All quality control was run during data collection. 

only define acronyms at first use from introduction onwards.

Response: We did as suggested 

Consent: In minors: who gave consent before data collection? Minors or guardians?

Response: Consent was obtained from parents/guardians whereas assent was obtained from minors as shown in the manuscript with track change. 

Results Did all participants answer all qustions? If not, n/N needs to be given for each variable.

% can be given before n (and put n in paranthases)

Response: The calculated sample size was 422; however, 416 (98.6%) were responded. The data presented in the result section reflect 416 participants. Every result is computed out of 416 respondents we have also presented the result as % (n/N) as shown in the manuscript with track change.

For the tables: consider not presenting in question format, but instead as reported HIV test, Has chewed khat, has smoked cigarettes, etc.

Response: We have corrected as suggested

Table 1: the variable monthly income is household or individual?

Response: it is monthly income of house hold as indicated in the manuscript with track change. 

Table 2: what does the p-value indicate? Difference between sexes? Include in the table title what it means.

Response: It was included based on the previous comment. Now, we have included in the title “…..among males and females…”

Again, make sure the acronyms for STIs have been presented previously.

Response: We have defined STIs in the introduction section. 

Factors associated with STIs: no need to reexplain methods.

Response: We omitted the subheading and modified the other subheading as ‘Seroprevalence of STI and its predictors’

´Grouped STs were associated with x, y and z or x, y and z were associated with increased odds of grouped STI – and make sure you´ve defined what grouped STIs are in methods)

Response: We have defined STI under operational definition 

Discussion Have a look again at what to include in the first paragraph of a discussion- you can include some overall riskfactors as well as STI results.

Response: We have included factors associated with STI in the first paragraph 

Lines starting 250: are these studies with reports of STI? Or findings in laboratory testing? It seems like as It is written that participants of these studies had to self-report their sti diagnosis (which is never going to be accurate)

Response: they are not self-report they are the findings of studies (reference)

Paragraph 3 needs to be restructured. Keep to one topic. The highest prevalence is HBV- what does that mean for vaccination? Further research should look at vaccination status.

Response: We think the reviewer is referring to paragraph 4. We presented them together because of their similarity in route of transmission and pathogenesis (HBV, HCV). 

All the paragraphs with comparisons from other LMIC can be grouped into one paragraph.

Response: Thank you for the comment. There are different ways of organizing paragraphs; we preferred to organize the discussion based on our objectives and nature of STIs. 

Make sure your study is comparable to what you present. What is given from Rhode Island doesn´t make sense (in incarcerated populations)

Response: We agree with the comment, we omitted it. 

Make sure your comparisons make sense. Your syphilis prevalence of 0.7% does to compare with Brazilian prevalence.

Response: We have corrected it as “….is low compared to….”

You should have a section on ´policy and program recommendations´ or at the very least be mentioning this throughout the discussion. A whole section would be better though.

Response: We have included heading ‘policy and program recommendation’

Limitations: use proper limitations and bias vocabulary https://jech.bmj.com/content/58/8/635

Response: We have modified the limitation somehow after refereeing to the link provided (Thank you)

Conclusions: you need to include program recommendations.

Response: we have included. 

Please define abbreviations within text as well (when you first use them)

Response: we dis as suggested. 

7. PLOS authors have the option to publish the peer review history of their article (what does this mean?). If published, this will include your full peer review and any attached files.

Do you want your identity to be public for this peer review? For information about this choice, including consent withdrawal, please see our Privacy Policy.

Reviewer #1: No

Reviewer #2: No

---

## [Editor Report · Decision Letter 2]

19 Dec 2022

Sero-prevalence and associated factors of sexually transmitted infections among youth-friendly services Attendees

PONE-D-22-12316R2

Dear Dr. Ali,

We’re pleased to inform you that your manuscript has been judged scientifically suitable for publication and will be formally accepted for publication once it meets all outstanding technical requirements.

Kind regards,

Chuanyi Ning, Ph.D.

Academic Editor

PLOS ONE
---

## [Editor Report · Acceptance letter]

12 Jan 2023

PONE-D-22-12316R2 

Sero-prevalence and associated factors of sexually transmitted infections among youth-friendly services Attendees 

Dear Dr. Ali:

I'm pleased to inform you that your manuscript has been deemed suitable for publication in PLOS ONE. Congratulations! Your manuscript is now with our production department. 

Kind regards, 

on behalf of

Dr. Chuanyi Ning 

Academic Editor

PLOS ONE